# Amorphous topological phases protected by continuous rotation symmetry

**Hélène Spring[1⋆], Anton R. Akhmerov[1] and Dániel Varjas[1,2,3†]**

**1** Kavli Institute of Nanoscience, Delft University of Technology,
P.O. Box 4056, 2600 GA Delft, The Netherlands
**2** QuTech, Delft University of Technology, P.O. Box 4056, 2600 GA Delft, The Netherlands
**3** Department of Physics, Stockholm University, AlbaNova University Center,
106 91 Stockholm, Sweden

⋆ helene.spring@outlook.com, † dvarjas@gmail.com
See also: online presentation recording.

## Abstract

Protection of topological surface states by reflection symmetry breaks down when the boundary of the sample is misaligned with one of the high symmetry planes of the crystal. We demonstrate that this limitation is removed in amorphous topological materials, where the Hamiltonian is invariant on average under reflection over any axis due to continuous rotation symmetry. We show that the edge remains protected from localization in the topological phase, and the local disorder caused by the amorphous structure results in critical scaling of the transport in the system. In order to classify such phases we perform a systematic search over all the possible symmetry classes in two dimensions and construct the example models realizing each of the proposed topological phases. Finally, we compute the topological invariant of these phases as an integral along a meridian of the spherical Brillouin zone of an amorphous Hamiltonian.

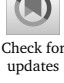

# 1   Introduction

Materials with a quasiparticle band gap in the bulk host protected edge states if they have a nontrivial topology. To determine whether an insulator or a superconductor is topological, one first determines the symmetry class of the quasiparticle Hamiltonian in this material, and then evaluates the topological invariant of the Hamiltonian's symmetry class [1, 2]. The topological invariant stays constant as long as the symmetry is preserved and the bulk stays gapped. While the specific properties of the surface states depend on details of the edge, they may not be removed by any symmetry-preserving surface perturbation due to the bulk-boundary correspondence.

The classification of topological phases started with the Altland-Zirnbauer classes, based on discrete onsite symmetries: particle-hole, time-reversal, and chiral symmetry [3, 4]. Topological crystalline phases were also classified [5–8], protected by crystal symmetries. The bulk-boundary correspondence, however, does not apply to all edges in this case: spatial symmetries such as reflection are broken by certain edge orientations [9] and the edge states may become gapped, as seen in the top panels of Fig. 1.

When perturbations are introduced to a system with nontrivial topology, the topological phases may be destroyed if the symmetries are affected. Perturbed symmetries present on average are able to provide topological protection [10]. Disordered systems that support topo-

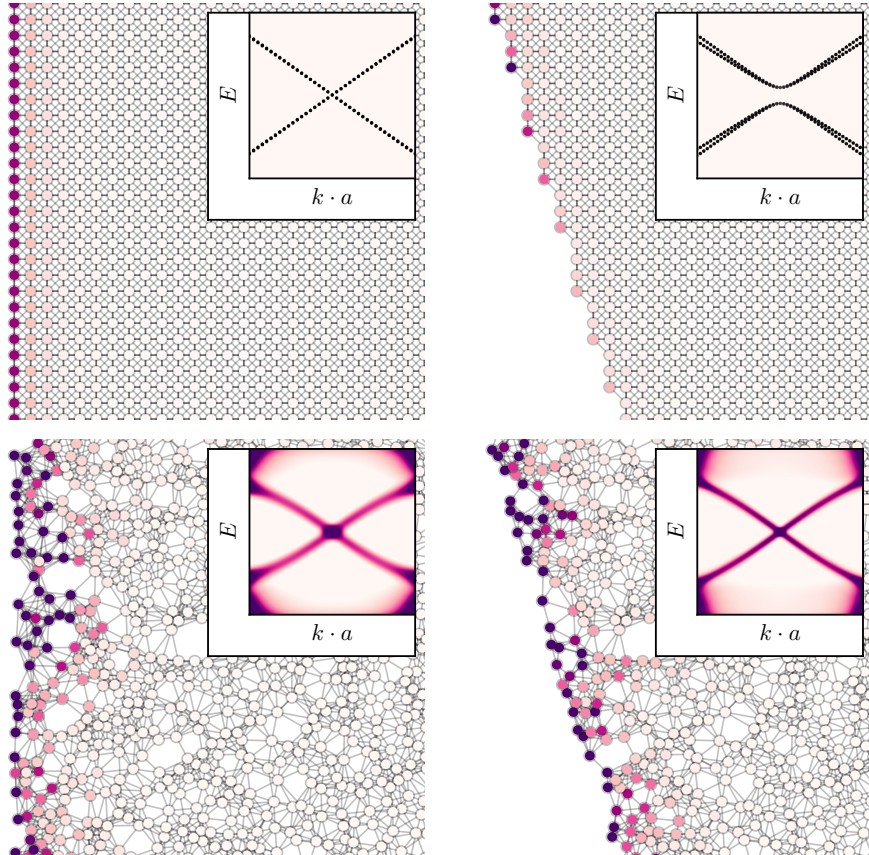

Figure 1: The zero-energy local density of propagating modes of the class D 8-band model in crystal and amorphous systems; darker site color indicates higher density. Insets: dispersion relation (top) and momentum-resolved spectral function (bottom) corresponding to straight and tilted edge terminations. The effective lattice constant of the amorphous system $a$ is given by $a = 1/\sqrt{\rho}$, where $\rho$ is the density of sites in the system. Plot details in App. A.

logical insulating phases with one exact symmetry and one or more average symmetries are called statistical topological insulators [11]. The surfaces of statistical topological insulators are delocalized and pinned to the midpoint of a topological phase transition, or critical point. A crystal surface that respects a crystalline symmetry on average is still able to host crystalline topological phases.

Unlike crystals, which break continuous rotation symmetry even on average, amorphous systems lack long-range order and are therefore on average compatible with continuous rotations. Strong topological, metallic and insulating phases as well as topological superconductivity have been studied in amorphous systems both theoretically [12–18] and experimentally [19–21].

In this work, we devise topological insulator (TI) phases in amorphous systems that rely on the presence of two average spatial symmetries: reflection symmetry and continuous rotation symmetry. The presence of both reflection symmetry and average continuous rotation symmetry promotes the protection of a crystalline topological phase to every edge orientation. We thus demonstrate that even though the topological phases presented here have crystalline or quasi-crystalline counterparts, only amorphous systems have guaranteed protection for all edge terminations. This study exposes the potential for realizing topological phases protected by average spatial symmetries that don't rely on macroscopic edge details.

The structure of the manuscript is as follows. In Sec. 2 we define the basic premise of spatial symmetries in amorphous systems. In Sec. 3 we study isotropic continuum systems and identify the symmetry groups containing reflection symmetry that protect gapless edge states. In Sec. 4 we construct amorphous tight-binding models, numerically demonstrate critical edge transport, and compare with a similar system on a regular square lattice. Finally, we formulate bulk topological invariants of our systems in Sec. 5. We conclude in Sec. 6 that amorphous models relying on spatial symmetries as well as one or more exact onsite symmetry to protect a topological phase are statistical topological insulators, provided the disorder of the amorphous system does not close the bulk gap.

## 2 Spatial symmetries in amorphous matter

Despite locally breaking all spatial symmetries, amorphous matter is generated by a highly symmetric ensemble of Hamiltonians. Specifically, the occurrence probability of any configuration is invariant under the action of any element of the Euclidean group. Furthermore, all structural correlations must decay sufficiently fast with distance. These conditions require care to satisfy and cannot be fulfilled by gradually moving sites from their crystalline positions. While there are several ways to simulate amorphous matter, we focus on tight-binding models defined on random graphs. The simplest way to create an amorphous array of site positions is choosing a sample of uncorrelated points in space. In order to reduce the fluctuations of the coordination number, we use a sphere-packing algorithm described in App. B instead.

The physics of amorphous systems obeys locality and homogeneity in the sense that the bulk Hamiltonian is generated by a local rule [22,23]. We require that the onsite and hopping terms only depend on the local environment: the configuration of atoms within a finite radius of the site or bond in question. For our toy models we take an even simpler case, where terms in the Hamiltonian only depend on the relative spatial positions of the orbitals:

$$\langle \mathbf{r}, n | \hat{H} | \mathbf{r}', m \rangle = H_{nm} \left( \mathbf{r} - \mathbf{r}' \right), \tag{1}$$

where $|\mathbf{r}, n\rangle$ is the $n$'th orbital on the site at position $\mathbf{r}$. While this restriction is not essential, it makes defining the models easier. Onsite terms have $\mathbf{r} - \mathbf{r}' \equiv \mathbf{d} = 0$, meaning all onsite terms in the bulk are identical. More generally, we allow $H(\mathbf{d})$ to be a random matrix whose distribution only depends on the hopping vector $\mathbf{d}$ to account for sources of disorder not captured by the underlying random graph or the simplified local rule. In this case we demand that the disordered ensemble is invariant under each spatial symmetry, whereas the onsite symmetries are obeyed exactly by each ensemble element.

An isotropic amorphous system has average continuous rotation symmetry under simultaneous rotation in spin and real space, meaning that terms in the Hamiltonian with a rotated local environment are related as:

$$U(\phi)H(\mathbf{d})U(\phi)^{-1} = H(R(\phi) \cdot \mathbf{d}), \tag{2}$$

with $U(\phi) = \exp(i\phi S_z)$, $S_z$ the onsite spin-$z$ operator, $R(\phi) = \exp(i\phi L_z)$, $L_z = \sigma_y$ the generator of two-dimensional real space rotations. Simultaneous invariance under continuous rotation and one reflection symmetry implies reflection invariance with any normal vector. The symmetry constraint imposed by a reflection operator with normal $\hat{\mathbf{n}}$ is:

$$U_{M_{\hat{\mathbf{n}}}} H(\mathbf{d}) U_{M_{\hat{\mathbf{n}}}}^{-1} = H(R_{M_{\hat{\mathbf{n}}}} \cdot \mathbf{d}), \tag{3}$$

where $R_{M_{\hat{\mathbf{n}}}} = \mathbb{1} - 2\hat{\mathbf{n}}\hat{\mathbf{n}}^T$ is the real space orthogonal action reversing the component in the $\hat{\mathbf{n}}$ direction. Commutation relations of $S_z$, $U_M$ and onsite symmetries are listed in App. C.

All previous considerations of this section apply to homogeneous and isotropic systems deep in the bulk. The vicinity of the edges of the system are, however, distinguishable from the bulk through the local environment, and have lower symmetry. Hence we allow the Hamiltonian to depend on the distance from the edge and the orientation of the edge. For example, near an infinite edge along the $y$ direction such that the system terminates for $x < 0$ we let

$$\langle \mathbf{r}, n | \hat{H} | \mathbf{r}', m \rangle = H^{\text{edge}}_{nm} \left( \mathbf{r} - \mathbf{r}', \hat{x} \cdot \frac{\mathbf{r} + \mathbf{r}'}{2} \right), \tag{4}$$

such that $\lim_{x \to \infty} H^{\text{edge}}(\mathbf{d}, x) = H(\mathbf{d})$. This local rule preserves average translation invariance along the edge, but may break the continuous rotation symmetry (2) of the bulk. A straight edge still preserves average reflection symmetry with normal parallel to the edge, so we demand that $H^{\text{edge}}$ satisfies (3) with fixed $x$ and $\hat{\mathbf{n}} = \hat{y}$.

## 3 Continuum systems

In the long wavelength limit an amorphous system is homogeneous and isotropic, resembling a continuum. We therefore start our analysis by studying continuum models with reflection and continuous rotation invariance. First we study the 1D edge theory to identify symmetry groups capable of protecting gapless edge modes. Next we construct 2D bulk models in these symmetry classes, and finally we demonstrate that straight domain walls host gapless modes as expected.

### 3.1 Symmetry groups protecting gapless edges

In order to find continuum models with gapless edges protected by reflection symmetry, we perform a systematic search of the Altland-Zirnbauer symmetry classes. For each class, we start with a minimal 1D Dirac Hamiltonian that respects the onsite symmetries. If mass terms are allowed in this Hamiltonian, *i.e.* it is trivial with only the onsite symmetries, we add a reflection symmetry. The Hamiltonian is a candidate model if the reflection symmetry protects the gapless edge by removing all mass terms.

Consider for example the edge of a class D system, the minimal two-band edge theory can always be written as $H_{\text{edge}}(k) = k\tau_x + m\tau_y$ with particle-hole symmetry acting as complex conjugation, $\mathcal{P} = \mathcal{K}$. In the absence of additional symmetries this model describes the edge of a trivial system because it is gapped for any nonzero $m$. Choosing a unitary reflection symmetry with $U_M = \tau_z$ the symmetry constraint $U_M H_{\text{edge}}(k) U_M^\dagger = H_{\text{edge}}(-k)$ forces $m = 0$. Hence this choice of reflection symmetry protects a single pair of counterpropagating gapless edge modes, and serves as a candidate for the edge theory of a topologically nontrivial bulk protected by reflection.

We perform the search of the Altland-Zirnbauer classes using the software package Qsymm [24]. In classes AII, DIII, CII and C the minimal model of a gappable edge is $4 \times 4$, in the rest of the classes it is $2 \times 2$. We fix a canonical form of the onsite symmetries, then vary the reflection-like symmetry using different products of Pauli matrices $\sigma$ and $\tau$ for its unitary part, also allowing it to act as an antiunitarity (with complex conjugation) and as antisymmetry (reversing the sign of the Hamiltonian). This approach tests every possible reflection-like symmetry up to basis transformations. In this basis, we have $U_M^2 = +\mathbb{1}$. The conventional fermionic reflection operator that obeys $U_M^2 = -\mathbb{1}$ is recovered by multiplying $U_M$ with $i$. This change of the overall phase does not affect the symmetry constraints on the Hamiltonian and only reverses commutation and anticommutation of $U_M$ with the antiunitary symmetries. For each choice of the symmetry group, we generate the most general $k$-linear Hamiltonian. If it

Table 1: Symmetry representations of 1D models where a unitary reflection symmetry $U_M$ protects gapless edges. $\sigma$ and $\tau$ are Pauli matrices. Only unitary-inequivalent symmetry representations are listed.

| Symmetry class | $U_M$ | $U_{\mathcal{P}}$ | $U_{\mathcal{T}}$ | $U_{\mathcal{C}}$ |
|---|---|---|---|---|
| AIII | $\tau_x$ | - | - | $\tau_x$ |
| BDI | $\tau_x$ | $\tau_0$ | $\tau_x$ | $\tau_x$ |
| D | $\tau_z$ | $\tau_0$ | - | - |
| DIII$_+$ | $\sigma_x\tau_z$ | $\sigma_0\tau_z$ | $i\sigma_z\tau_y$ | $\sigma_z\tau_x$ |
| DIII$_-$ | $\sigma_z\tau_x$ | | | |
| CII | $\sigma_y\tau_y$ | $i\sigma_y\tau_0$ | $i\sigma_0\tau_y$ | $\sigma_y\tau_y$ |

does not contain $k$-independent mass terms capable of opening a gap at half-filling, we note it as a candidate. When presenting the results in Table 1 we only list one representative of various reflection operators related by unitary basis transformations. In the rest of the manuscript we focus on the more natural symmetry groups with unitary reflection symmetry, see App. D for symmetry groups with reflection antisymmetries.

Because we are searching for phases whose surfaces are driven to a critical point by spatial disorder, we expect to find protected gapless phases in the presence of strong disorder in symmetry classes that host nontrivial topological phases in 1D. This requires the disorder to respect all non-spatial symmetries in a given class exactly, and the spatial symmetries on average [11]. In this case the additional reflection symmetry forces the edge to the critical point of a topological phase transition. The result of our search confirms this expectation, we find unitary reflection symmetries in classes AIII, BDI, CII, D and DIII. We observe that in all the chiral classes $[U_M, \mathcal{C}] = 0$, and in all cases $[U_M, \mathcal{P}] = [U_M, \mathcal{T}] = 0$ except for one of the choices for class DIII where $\{U_M, \mathcal{P}\} = \{U_M, \mathcal{T}\} = 0$. We denote the case with commuting reflection DIII$_+$ and the case with anticommuting reflection DIII$_-$ in the following.

When attempting to extend these symmetries to the 2D bulk, we find that these symmetry representations do not admit a consistent continuous rotation symmetry with $S_z = \pm 1/2$ (see App. C) in a way that allows a gapped bulk, so we double the Hilbert-space. We perform a systematic search for symmetry representations by taking the tensor product of each edge symmetry operator with a Pauli matrix, taking $S_z$ as $1/2$ times the product of Pauli matrices and ensuring that the appropriate commutation relations are maintained. While this search is not exhaustive, it produces gapped bulk models realizing all the edge symmetry classes. The exact forms of the onsite and spatial symmetries in the bulk are listed in App. D.

## 3.2 Bulk models

We use Qsymm to obtain continuum models in reciprocal space ($k$-space) compatible with the bulk symmetry representations found in the previous subsection. The symmetry constraints have the following form in $k$-space:

$$U(\phi)H(\mathbf{k})U(\phi)^{-1} = H(R(\phi) \cdot \mathbf{k}), \tag{5}$$

$$U_M H(\mathbf{k})U_M^{-1} = H(R_M \cdot \mathbf{k}), \tag{6}$$

$$U_{\mathcal{C}} H(\mathbf{k})U_{\mathcal{C}}^{-1} = -H(\mathbf{k}), \tag{7}$$

$$U_{\mathcal{P}} H^*(\mathbf{k})U_{\mathcal{P}}^{-1} = -H(-\mathbf{k}), \tag{8}$$

$$U_{\mathcal{T}} H^*(\mathbf{k})U_{\mathcal{T}}^{-1} = H(-\mathbf{k}). \tag{9}$$

We generate all symmetry allowed terms up to linear order in $k$ in 4-band models for classes AIII, BDI and D, and 8-band models in classes DIII and CII. We also include one $\mathbf{k}^2$ term to

ensure proper regularization in the large $k$ limit (see Sec. 5.1). We split the Hamiltonian into $k$-independent onsite (or mass) terms and $k$-dependent hopping terms as $H(\mathbf{k}) = H^{\text{os}} + H^{\text{hop}}(\mathbf{k})$, see the explicit enumeration of all the terms in App. E.1.

For classes AIII, BDI and D, while the minimal 4-band models have gapped bulk, we find that these systems are non-generic for the prescribed symmetries. The minimal class BDI model consists of two decoupled blocks resulting in an additional onsite unitary symmetry, the class AIII model has an additional time-reversal symmetry, and the class D model remains decoupled at $\mathbf{k} = 0$ resulting in extra protection for the edge modes. To get rid of the additional symmetries, we consider a doubled Hamiltonian:

$$H_{8\times8}(\mathbf{k}) = \begin{pmatrix} H(\mathbf{k}) & H^{\text{c}}(\mathbf{k}) \\ H^{\text{c}}(\mathbf{k})^{\dagger} & H'(\mathbf{k}) \end{pmatrix}, \tag{10}$$

where $H$ is topological, $H'$ is trivial, and $H^{\text{c}}$ is weak. The forms of the coupling between the two copies, $H^{\text{c}}$, are listed in App. E.1. We then confirm that the resulting doubled model remains topological, and the additional symmetries are removed. The 8-band CII and DIII models have no unwanted symmetries, so they are not doubled.

## 3.3 Gapless domain wall modes

To show that the bulk models have the expected edge physics, we obtain the continuum edge spectra of our models by considering an infinite 2D system with a domain wall. We assign a spatial dependence to the chemical potential, such that at $x = 0$ its sign is flipped, making the system topological for $x > 0$ and trivial for $x < 0$. Topological edge modes are confined to the interface and decay exponentially into the bulk.

The continuum model $H_{\text{cont}}(k)$ is obtained from (10) by replacing $k_y$ with a free parameter $k$ and $k_x$ with its real-space form $-i\partial_x$. We cast the eigenvalue problem $H_{\text{cont}}\Psi = E\Psi$ into the form of a system of linear differential equations $A(k)\partial_x\Psi + B(k, x, E)\Psi = 0$. We find all the solutions on the left and right side of the domain wall separately, using the ansatz

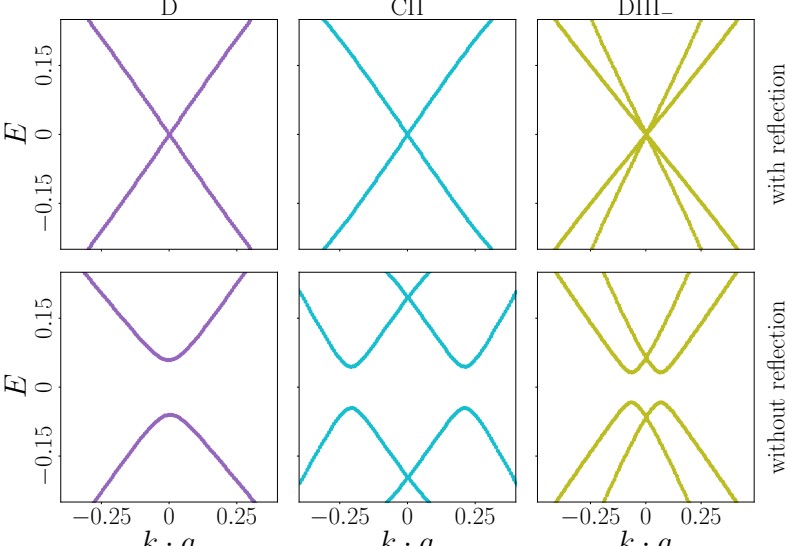

Figure 2: Domain wall spectra of the continuum models in classes D, CII and class DIII obtained numerically. For class DIII, the anticommuting case DIII_ is represented. With reflection symmetry the boundary spectrum is gapless (top row), while reflection-breaking terms open a gap (bottom row).

$\Psi_{\text{L/R}}(x) = \psi_{\text{L/R}} \exp(-\lambda_{\text{L/R}}|x|)$ to obtain $(A - \lambda_{\text{L/R}}B)\psi_{\text{L/R}} = 0$. We solve this generalized eigenvalue problem and concatenate the solutions for $\psi^i_{\text{L/R}}$ into a single matrix $W$. A global solution needs to be continuous at $x = 0$, and it exists if there is a nonzero linear combination of the left mode vectors $\psi^i_{\text{L}}$ that is also a linear combination of right mode vectors $\psi^i_{\text{R}}$. We therefore obtain the edge spectrum by numerically finding points in the $(E, k)$ plane where $W$ is singular [25].

This analysis shows that all the continuum models we consider have gapless modes at the boundary between topologically trivial and non-trivial regions protected by mirror symmetry, as shown in Fig. 2. Any perturbation that breaks the reflection symmetry opens a gap, even if it preserves all the onsite symmetries. The class D spectrum is representative of the AIII and BDI spectra. The edge modes of the CII model are doubly degenerate due to the combination of its reflection and time-reversal symmetries.

## 4 Amorphous systems

In this section we demote the exact spatial symmetries of the continuum models to average symmetries by using tight-binding Hamiltonians on an amorphous graph, and demonstrate that the topological protection by reflection and continuous rotation symmetry persists.

### 4.1 Amorphous tight-binding Hamiltonians

In order to extract the scaling behaviour of the edges of an amorphous system, we construct real space tight-binding models using the symmetry considerations outlined in Sec. 2. While the problem formally looks very similar to the $k$-space case replacing $\mathbf{k}$ with $\mathbf{d}$, onsite symmetries behave differently in real space:

$$U_{\mathcal{C}}H(\mathbf{d})U_{\mathcal{C}}^{-1} = -H(\mathbf{d}), \tag{11}$$

$$U_{\mathcal{P}}H^*(\mathbf{d})U_{\mathcal{P}}^{-1} = -H(\mathbf{d}), \tag{12}$$

$$U_{\mathcal{T}}H^*(\mathbf{d})U_{\mathcal{T}}^{-1} = H(\mathbf{d}). \tag{13}$$

Hermitian adjoint reverses hoppings, so $H(\mathbf{d})$ is generally nonhermitian, but obeys a modified hermiticity condition:

$$H(-\mathbf{d}) = H(\mathbf{d})^{\dagger}. \tag{14}$$

With these modifications, we use Qsymm to generate all symmetry-allowed hopping terms $H^{\text{hop}}(\hat{\mathbf{d}})$ as first order polynomials of the components of $\hat{\mathbf{d}}$. The hopping terms obtained in this way have a sufficiently general dependence on the bond direction for our purposes. The onsite terms obey the same symmetry conditions as in $k$-space, so we use the same $H^{\text{os}}$ as in the previous section. In order to make the Hamiltonian short-ranged without changing its symmetries, we make the hoppings decay exponentially with bond length, see App. F. Again we consider doubled models in classes AIII, BDI and D, the results are listed in App. E.2.

### 4.2 Transport properties of the amorphous edge

To demonstrate that our amorphous systems are statistical topological insulators, we show that their transport signatures match those of 1D disordered systems at the critical point of a topological phase transition. The transmission amplitudes $t_i$ are random variables that depend on the disorder configuration of the system and the conductivity is given by $g = \sum_i |t_i|^2$ [26]. At the critical point the transmission amplitude distribution universally obeys $\alpha = \text{arccosh}(1/|t|)$ such that $\alpha$ has half-normal distribution with scale parameter $\sigma$ that grows with the edge

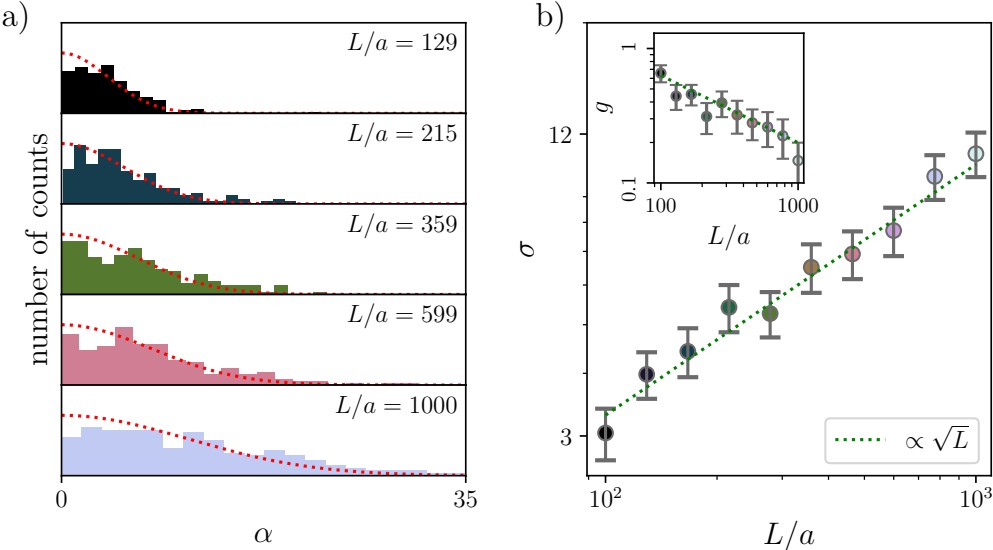

Figure 3: Critical transport scaling for the 8-band class D amorphous system with onsite disorder. a) histogram of $\alpha$ for various system sizes $L$ from 93 different amorphous system realizations. In red: maximum likelihood estimate fit of a half-normal distribution to the data. b) length dependence of $\sigma$, the scale parameter of the half-normal fits. Inset: average conductance $g$ as a function of system size. The dashed lines are the $L^{\pm 1/2}$ fit to the scaling data.

length $L$ as $\sigma \propto \sqrt{L}$ [27, 28]. The resulting disorder-averaged conductance has power-law decay $g \propto L^{-1/2}$.

We fit the $\alpha_i$ obtained from numerical transport calculations on edges of the class D amorphous model with various edge lengths for several random realizations of the amorphous system to half-normal distributions (see App. B). The top panel of Fig. 3 shows the histograms of $\alpha$, and the bottom panel shows that we recover the relation $\sigma \propto \sqrt{L}$ for the standard deviation of $\alpha$ and $g \propto L^{-1/2}$ for the conductance. Here we use a model with Gaussian distributed onsite disorder only respecting particle-hole symmetry to show the critical scaling of the conductance $g$. We expect that allowing the onsite terms to depend on the local environment, as is the case for more detailed models of amorphous matter, would have a similar effect. While we recover the scaling of $\sigma$ without onsite disorder, we find that the intrinsic disorder from the underlying random graph is too weak to detect the conductance scaling at numerically feasible system sizes, see App. G.

## 4.3 Analogous model on the square lattice

The way we defined our hopping Hamiltonians allows us to use them on any graph, including regular crystal lattices. This lets us demonstrate that breaking the rotation and reflection symmetries to a discrete subgroup opens a gap on reflection asymmetric edges. We calculate the band structures of periodic crystal strips whose edges are terminated along different directions and inspect the dispersion of the edge modes spanning the bulk gap.

Using a sufficiently general model on the square lattice that breaks all additional symmetries beyond the onsite and spatial symmetries we prescribe (see App. F) we find that reflection-breaking edges on the square lattice are gapped. Fig. 4 compares edges oriented along $[1,0]$ and $[2,1]$, in the first case reflection symmetry of the edge protects gapless modes, while in the second case it does not.

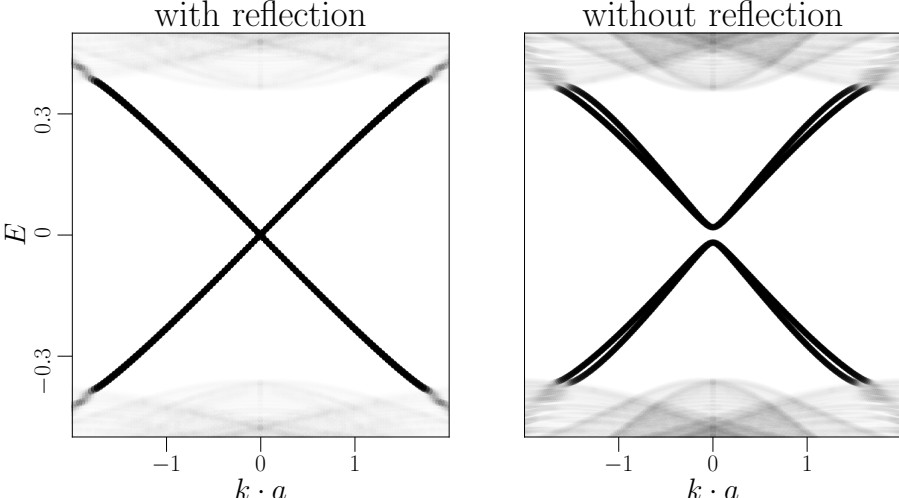

Figure 4: Band structures of the class D model on periodic crystal strips for different edge terminations and distance dependences. The left panel shows bands along the reflection symmetric [1 0] edge, and the right panel shows bands along the [2 1] edge, that breaks reflection symmetry. Transparency of the dispersion bands is directly proportional to their participation ratio. Plot details in App. A and App. F.

# 5 Bulk invariant

We have demonstrated the robustness of gapless edge modes protected by reflection symmetry in both continuum and amorphous systems. In this section we give an explicit invariant characterizing the topological phase without referring to edge properties.

## 5.1 Continuum models

We construct the 2D bulk invariants of the rotation symmetric continuum Hamiltonians from the 1D invariants of the same symmetry class. This is motivated by the fact that the Hamiltonian on any 1D line in $k$-space specifies the Hamiltonian everywhere in the 2D $k$-space through rotation symmetry. To relate to 1D invariants defined on a finite Brillouin zone, we require the Hamiltonian to be sufficiently regularized: the eigenvectors of $H(\mathbf{k})$ must become independent of the direction of $\mathbf{k}$ for the limit $|\mathbf{k}| \to \infty$. For example, the quadratic terms of (29) dominate the $k$-space Hamiltonian in this limit, making it insensitive to the signs of $k_x$ and $k_y$. This allows compactification of the $\mathbb{R}^2$ momentum space of the continuum to a sphere $S^2$ by identifying all infinitely far points to a single point, which we denote $\mathbf{k} = \infty$. We use a stereographic projection to construct this mapping from $\mathbb{R}^2$ to $S^2$. The Hamiltonian at $\mathbf{k} = \mathbf{0}, \infty$ is invariant under continuous rotations [29, 30] as well as under all reflection symmetries. Furthermore, the Hamiltonian on any line connecting these two points determines the Hamiltonian everywhere on the $k$-space sphere. Therefore it is natural to think of the momentum space of an amorphous material as a spherical Brillouin zone with North and South poles at $\mathbf{k} = \mathbf{0}, \infty$, an axis of rotation along the $\hat{z}$ axis, and mirror lines on every meridian.

The invariant in 1D class D is $\nu_{D1} = \text{sign}[\text{pf}\, H(k=0) \cdot \text{pf}\, H(k=\pi)]$ where pf denotes the Pfaffian and $H(k) = -H(k)^*$ is the class D Hamiltonian in the Majorana basis. This generalizes to the 2D continuum as $\nu_{D2} = \text{sign}[\text{pf}\, H(\mathbf{0}) \cdot \text{pf}\, H(\infty)]$. This invariant, however, is only nontrivial if the system has nonzero Chern number, because $\exp(i\pi C) = \nu_{D2}$ [23], which is not possible with mirror symmetry. To define a new invariant in the presence of a unitary mirror symmetry whose eigenvalues are invariant under particle-hole conjugation ($U_M \mathcal{P} = \mathcal{P} U_M$ for

Table 2: Classification of topological phases in continuum and amorphous systems protected by continuous rotation and unitary reflection symmetry. The classification does not include the strong 2D invariant that is an independent $\mathbb{Z}_2$ invariant in class $\text{DIII}_-$. In all other classes reflection symmetry enforces a trivial strong invariant. For details on how disorder leads to the distinction between the continuum and amorphous classification, see App. H.

| Symmetry class | continuum | amorphous |
|:---:|:---:|:---:|
| AIII | $\mathbb{Z}$ | $\mathbb{Z}_2$ |
| BDI | $\mathbb{Z}$ | $\mathbb{Z}_2$ |
| D | $\mathbb{Z}_2$ | $\mathbb{Z}_2$ |
| $\text{DIII}_+$ | $\mathbb{Z}_2$ | $\mathbb{Z}_2$ |
| $\text{DIII}_-$ | $2\mathbb{Z}$ | $0$ |
| CII | $2\mathbb{Z}$ | $2\mathbb{Z}_2$ |

$U_M^2 = +\mathbb{1}$, as is the case for the model studied in the manuscript) we apply the above formula to the two reflection sectors separately:

$$\nu_{\text{M}} = \text{sign}\left[\text{pf}\,H_{\pm}(\mathbf{0}) \cdot \text{pf}\,H_{\pm}(\boldsymbol{\infty})\right], \tag{15}$$

where $H_{\pm}$ is the Hamiltonian restricted to the $\pm 1$ eigensubspace of $U_M$. The choice of the reflection sector is arbitrary, as the product of the invariants for the two sectors equals $\nu_{\text{D2}} = +1$.

To prove that a nontrivial bulk invariant corresponds to gapless edge states, we consider a system with a straight edge in the $y$ direction preserving $M_y$. Restricting to zero momentum along the edge ($k_y = 0$) we get a half-infinite 1D system, whose bulk is described by $H(k_x, 0)$ that is invariant under $M_y$ for every $k_x$. The bulk invariant derived above is exactly the reflection-resolved strong invariant of the 1D system, indicating zero modes at a real space boundary for each mirror sector in the nontrivial phase. These zero modes correspond to the crossing of the edge modes at $k_y = 0$.

To construct the topological invariant in other symmetry classes, we follow a similar procedure. The topological invariants of odd-dimensional systems with chiral symmetry are winding numbers [5]. Therefore, the bulk invariants of the AIII, BDI, and CII classes is the winding number of a single reflection sector modulo 2. In class $\text{DIII}_+$ we construct a reflection-resolved $\mathbb{Z}_2$ invariant analogous to the class DIII Pfaffian invariant. We summarize the resulting classification of topological phases protected by unitary reflection and continuous rotation symmetry in continuum and amorphous systems in Table 2. Because the topological invariant is an integral along a high-symmetry line in $k$-space, these expressions coincide with the topological invariants of reflection-protected phases in crystalline materials [31–33].

## 5.2 Effective Hamiltonian of amorphous models

Without translation invariance it is still possible to detect the bulk gap closings that accompany topological phase transitions through the density of states $\rho(E) = N^{-1}\,\text{tr}\,\delta(\hat{H} - E)$ of a large finite system with $N$ sites. Fig. 5 (a) shows the density of states of the class D amorphous model as the chemical potential $\mu$ is tuned across two phase transitions. We observe two bulk gap closings, and a small constant density of states in the bulk gap due to edge states in the topological phase. To gain even more insight, we introduce the momentum-resolved spectral function

$$A(\mathbf{k}, E) = \sum_n \langle \mathbf{k}, n | \delta(\hat{H} - E) | \mathbf{k}, n \rangle, \text{ with } \langle \mathbf{r}, n | \mathbf{k}, m \rangle = N^{-1/2} \exp(i\mathbf{k}\mathbf{r})\delta_{nm}, \tag{16}$$

so that $|\mathbf{k}, n\rangle$ is a plane-wave state localized in the $n$'th orbital. We use the spectral function with momentum parallel to the edge to detect edge states in finite samples, as shown in Fig. 1. It is also well defined in the $\mathbf{k} \to \infty$ limit: because our amorphous samples are isotropic and the sites are always separated by a finite distance (see App. B), the relative phase on each bond in the plane wave converges to a uniform independent random phase. Fig. 5 (b) and (c) show that the two gap closings observed earlier are different: one occurs at $\mathbf{k} = \mathbf{0}$ and the other at $\mathbf{k} = \infty$.

In order to apply the construction of bulk invariants to amorphous systems, we introduce the effective $k$-space Hamiltonian [17, 23] $H_{\text{eff}}(\mathbf{k}) = G_{\text{eff}}(\mathbf{k})^{-1}$ through the projection of the single-particle Green's function onto plane-wave states:

$$G_{\text{eff}}(\mathbf{k})_{m,n} = \langle \mathbf{k}, m| \hat{G} |\mathbf{k}, n\rangle , \tag{17}$$

where $\hat{G} = \lim_{\eta \to 0}(\hat{H} + i\eta)^{-1}$ is the Green's function of the full real space Hamiltonian $\hat{H}$. Fig. 5 shows the relation to $A(\mathbf{k}, E)$. The spectrum of $H_{\text{eff}}(\mathbf{k})$ closely follows the peaks of the spectral function, especially near the gap closing points. The key properties of $H_{\text{eff}}$ are that it transforms the same way under symmetries as continuum Hamiltonians discussed before, its gap only closes when the gap in the bulk $\hat{H}$ closes [23], and it is properly regularized in the $\mathbf{k} \to \infty$ limit [17]. Hence, the bulk invariants defined for continuum systems are directly applicable to detecting topological phase transitions in amorphous systems. We show in Fig. 5 (d) for the class D amorphous model that the bulk invariant is non-trivial ($\nu_M = -1$) for intermediate values of the chemical potential.

# 6 Conclusions and Discussion

We introduced statistical topological insulator phases in two-dimensional amorphous systems that rely on average spatial symmetries for protection. We demonstrated that in the non-trivial phase the edge behaves as a 1D critical system of the same symmetry class by observing power-law scaling of the transport properties. We found topological invariants characterizing the bulk, and showed that the critical edge physics is not a result of fine-tuning, but is protected by the average reflection symmetry that is present on all straight edges of amorphous samples.

Comparing our results to similar work on higher-order topological insulators in quasicrystals protected by eight and twelvefold rotation symmetry [23, 34, 35] raises a natural question: can the amorphous phases protected by continuous rotation symmetry be described as a limit of systems with increasingly fine discrete rotation symmetry? It also remains an open question how to extend the topological classification to materials with multiple atom species.

Superconductivity is known to exist in amorphous thin films [36]. In the cases where we found new amorphous topological phases, however, the reflection symmetry commutes with time-reversal and particle-hole symmetry, while the physical reflection symmetry of $s$-wave superconductors anticommutes with onsite unitary symmetries. Hence condensed matter realizations of these symmetry classes are only feasible in the presence of reflection-odd (e.g. $p$-wave) pairing. It is possible that favourable energetics can result in an effective chiral symmetry, but such materials would be highly fine-tuned. Shiba glass systems consisting of atoms randomly deposited on surfaces have also been proposed as a platform for two-dimensional amorphous topological superconductivity [14]. Engineered systems, so called "topological simulators", can serve as an experimental demonstration of the phenomena studied in this work: the amorphous class BDI model could be naturally realized in disordered acoustic and mechanical meta-materials [37–39], while the other symmetry classes may be realized in a variety of systems including ultracold atoms [40], photonic crystals [41, 42], or coupled electronic circuit elements [43].

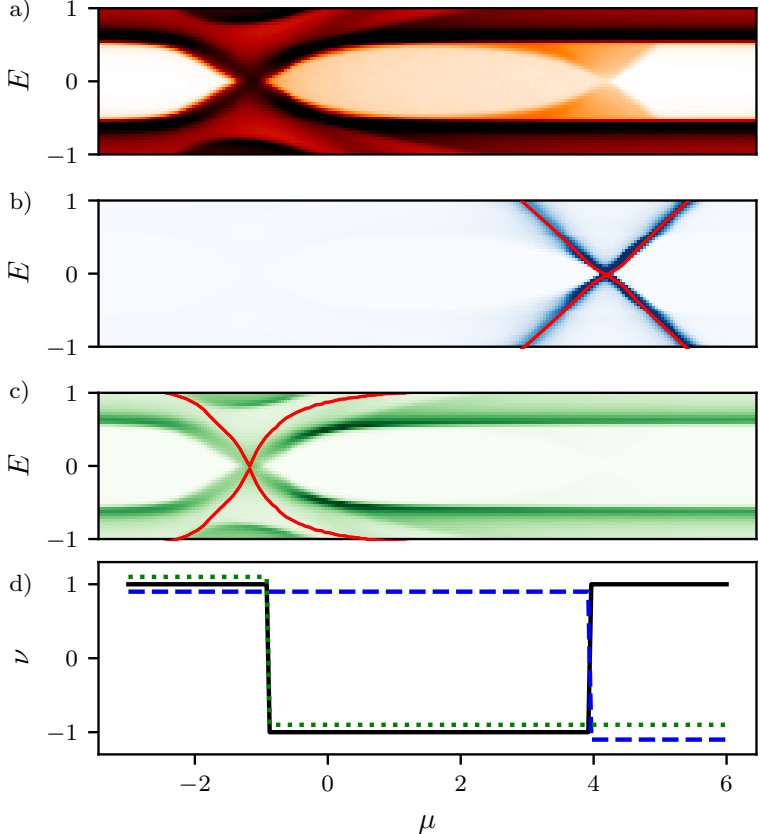

Figure 5: Topological phase transitions of the amorphous class D model as a function of the chemical potential. (a) Density of states of a finite amorphous sample. Darker color indicates higher density. (b) Spectral function at $\mathbf{k} = \mathbf{0}$ of a finite amorphous sample. The spectrum of the effective Hamiltonian is overlaid in red. (c) Same as (b) but at $\mathbf{k} = \boldsymbol{\infty}$. (d) Topological invariant $\nu_M$ (solid line). The dashed and dotted lines correspond to $\mathrm{sign}[\mathrm{pf}H_+^{\mathrm{eff}}(\mathbf{0})]$ and $\mathrm{sign}[\mathrm{pf}H_+^{\mathrm{eff}}(\boldsymbol{\infty})]$ respectively, offset along the vertical axis for visual clarity.

Our findings pave the way for a new classification of amorphous systems. Because the symmetry groups generated by continuous rotations are non-abelian in dimensions $d > 2$, we expect even richer topological classification in higher dimensions.

## Data availability

The data shown in the figures, as well as the code generating all of the data is available at [44].

## Author contributions

The initial project goal was formulated by D. V. and A. A. and was later refined with contributions from all authors. D. V. and A. A. supervised the project. H. S. performed the numerical and computer algebra calculations with assistance from all authors. D. V. formulated the bulk invariants. H. S. wrote the initial draft of the manuscript. All authors discussed the results and contributed to writing the manuscript.

## Acknowledgments

The authors thank I. C. Fulga, A. G. Grushin, F. Hassler and K. Pöyhönen for useful discussions. D. V. was supported by NWO VIDI grant 680-47-53, the Swedish Research Council (VR) and the Knut and Alice Wallenberg Foundation. A. A. and H. S. were supported by NWO VIDI grant 016.Vidi.189.180.

## A   Model and plotting parameters

In this section additional details of the plots are listed, if any, in order of appearance.

For Fig. 1, $f$ from (45) is set to 0.2 for $o_1$ and $o_4$ of (42). The data was obtained for systems containing 2500 sites.

The bottom panels of Fig. 2 are obtained by adding mirror-breaking terms to the continuum Hamiltonian models.

Fig. 3 is obtained from the class D model with added Gaussian noise terms that conserve particle-hole symmetry exactly. The amplitude of the noise terms $\gamma_i$ is $\frac{\gamma_i}{\mu} = 0.3 * x_i$ with $x_i$ a random number from a normal distribution with mean 0 and standard deviation 1, and $\mu$ the chemical potential of the topological sector of the model. The number of sites in the system vary from 5000 to 50000.

The data presented in Fig. 4 and 6 is obtained with $f = 0.2$ or $f = 1$ (as indicated) for the hopping terms $o_1$ and $o_4$ of (42). The periodic strips all have a width of 100 sites in the non-periodic direction.

Fig. 5 was obtained from a system with 40000 sites.

Fig. 7 is obtained with $f = 1.5$ for the hopping terms $t$ and $d$ of (37) of the non-trivial and trivial sectors of the AIII model respectively, and $o_4$ of (38). The class BDI data is obtained with $f = 0.7$ for $t$ of (39) of the non-trivial sector, and $o_2$ from (40). The class CII data is obtained with $f = 0.7$ for $t_1$ and $t_4$ of (43). The class DIII data is obtained with $f = 2$ for $o_1$ and $o_4$ of (44). The periodic strips all have a width of 100 sites in the non-periodic direction.

Fig. 8 was obtained from the class D model by setting $f = 0.7$ for hopping terms $t_1$, $d_2$ and $o_4$ of (41) and (42). The number of sites in the system vary from 5000 to 50000.

Fig. 9 was obtained from systems with 100 sites and Fig. 10 was obtained from systems with 2500 sites.

## B   Numerical methods

In the numerical calculations we use hard-disk amorphous structures [21]. To generate a structure, we randomly add atomic sites in a fixed volume from an uncorrelated uniform distribution. Treating atoms as hard disks, we reject new sites closer than a fixed distance to existing sites, and this procedure is performed until the goal density is reached. This procedure reduces density fluctuations and avoids sites that are very close to each other, matching the distance distribution function of a realistic amorphous system more closely than independent uniformly distributed points. We include hopping terms in the Hamiltonian for bonds connecting each site to a maximum number of $N$ neighbours falling within a maximum bond length $R$. The values of $N$ and $R$ are chosen such that the exponentially decaying hopping amplitudes to further neighbours can be safely neglected, resulting in a sparse Hamiltonian.

We use the software package Kwant [45] to generate the lattice Hamiltonians and for transport calculations. The transmission eigenvalues are obtained via the calculation of the scattering matrix using Kwant. The transmission amplitudes $t_i$ are given by the singular values

Table 3: Symmetry representations of 1D models where a reflection antisymmetry (that anticommutes with the Hamiltonian) with unitary part $U_M$ protects gapless edges. $\sigma$ and $\tau$ are Pauli matrices. Only unitary-inequivalent symmetry representations are listed.

| Symmetry class | $U_M$ | $U_{\mathcal{P}}$ | $U_{\mathcal{T}}$ |
|---|---|---|---|
| A | $\tau_0$ | - | - |
| AI | $\tau_0$ | - | $\tau_x$ |
| D | $\tau_0$ | $\tau_0$ | - |
| AII | $\sigma_0\tau_0$ | - | $\sigma_0\tau_y$ |
| C | $\sigma_0\tau_0$ | $\sigma_0\tau_y$ | 0 |

of the transmission block of the scattering matrix. Pfaffians are calculated using Pfapack [46]. The numerical density of states, momentum-resolved spectral function, and effective Hamiltonian calculations are performed using the kernel polynomial method [17, 23, 47, 48].

# C  Commutation relations of the symmetry operators

In real space, conjugating a rotation with a mirror results in a rotation in the opposite direction:

$$MR(\phi)M^{-1} = R(-\phi). \tag{18}$$

Demanding that there are no nontrivial onsite unitary symmetries, this implies for the unitary parts that

$$U_M e^{i\phi S_z} U_M^{-1} e^{i\phi S_z} = e^{i\alpha(\phi)}\mathbb{1}. \tag{19}$$

Differentiating with respect to $\phi$ and setting $\phi = 0$ yields

$$U_M S_z U_M^{-1} = -S_z + \alpha'\mathbb{1}, \tag{20}$$

where $\alpha' = d\alpha/d\phi|_{\phi=0}$. As the spectra of the two sides need to be equal, and the spectrum of $S_z$ consists of only integer or half-integer values, we find that $\alpha' \in \mathbb{Z}$. Redefining $S_z \to S_z - (\alpha'/2)\mathbb{1}$ the symmetry constraint on the Hamiltonian does not change, and we find that $S_z$ and $U_M$ anticommute. This also implies that the spectrum of $S_z$ is symmetric and $\text{tr}\, S_z = 0$, which is also a sufficient condition for the anticommutation with $U_M$, hence we assume $\text{tr}\, S_z = 0$ in the rest of the manuscript without loss of generality. Similar calculation shows that discrete onsite antiunitary (anti)symmetries (particle-hole and time-reversal) anticommute with $S_z$, and chiral symmetry commutes with $S_z$ in the absence of unitary symmetries.

# D  Details of symmetry representations

Besides the unitary mirror symmetries listed in the main text, we find several cases where a reflection antisymmetry (an operator that reverses $k$ and the energy) protects gapless edge states in continuum models. Since combinations of the reflection-like symmetry with any of the onsite symmetries is also a reflection-like symmetry providing the same protection, we omit such repetitions when listing the results in Table 3. We consider the results in classes A, AI and AII an artefact of using continuum models with perfect translation invariance, and expect that these are not viable for an amorphous system since they localize in the presence of disorder that makes the reflection antisymmetry only an average symmetry [49].

Table 4: Symmetry representations of 2D bulk models with unitary reflection and rotation symmetry. $\rho$, $\sigma$ and $\tau$ are Pauli matrices. The chemical potential terms are $\mu\sigma_z\tau_z$ for the 4-band models, $\mu\rho_z\sigma_z\tau_0$ for CII and $\mu\rho_z\sigma_0\tau_z$ for DIII.

| Symmetry class | $U_M$ | $U_\mathcal{P}$ | $U_\mathcal{T}$ | $U_\mathcal{C}$ | $S_z$ |
|---|---|---|---|---|---|
| AIII | $\sigma_x\tau_y$ | - | - | $\sigma_x\tau_0$ | $\frac{1}{2}\sigma_0\tau_z$ |
| BDI | $\sigma_x\tau_x$ | $\sigma_0\tau_x$ | $\sigma_x\tau_x$ | $\sigma_x\tau_0$ | $\frac{1}{2}\sigma_0\tau_z$ |
| CII | $\rho_y\sigma_y\tau_y$ | $i\rho_z\sigma_y\tau_0$ | $i\rho_z\sigma_0\tau_y$ | $\rho_0\sigma_y\tau_y$ | $\frac{1}{2}\rho_z\sigma_0\tau_y$ |
| D | $\sigma_x\tau_x$ | $\sigma_0\tau_x$ | - | - | $\frac{1}{2}\sigma_0\tau_z$ |
| DIII$_-$ | $\rho_z\sigma_x\tau_z$ | $\rho_x\sigma_0\tau_z$ | $i\rho_x\sigma_z\tau_y$ | $\rho_0\sigma_z\tau_x$ | $\frac{1}{2}\rho_0\sigma_y\tau_z$ |

The result of the search for 2D symmetry representations compatible with the edge symmetries is not unique: we pick one of several unitary equivalent choices for each Altland-Zirnbauer symmetry class. The specific forms of the symmetry representations that define the models in App. E are listed in Table 4.

For the 4-band models, we define the basis space of the unitary parts of the symmetry operators as the direct product $\sigma\otimes\tau$, with $\sigma$ and $\tau$ as Pauli matrices in sublattice and spin space respectively, such that the chemical potential terms of the models are $\mu\sigma_z\tau_z$. For the 8-band models, the basis space is extended to $\rho\otimes\sigma\otimes\tau$, where $\rho$ is also a Pauli matrix. For the doubled AIII, BDI and D models we extend the symmetries by multiplying with $\rho_0 = \mathbb{1}_2$.

# E  Model Hamiltonians

## E.1  Continuum Hamiltonians

The onsite Hamiltonians in both the continuum and amorphous bulk models are given by:

$$H^{\text{os}}_{\text{AIII}} = \mu\sigma_z\tau_z + \lambda\sigma_y\tau_z \tag{21}$$

$$H^{\text{os,c}}_{\text{AIII}} = \lambda_1\sigma_z\tau_z + i\lambda_2\sigma_y\tau_z \tag{22}$$

$$H^{\text{os}}_{\text{BDI}} = \mu\sigma_z\tau_z \tag{23}$$

$$H^{\text{os,c}}_{\text{BDI}} = \lambda_1\sigma_z\tau_z + i\lambda_2\sigma_y\tau_z \tag{24}$$

$$H^{\text{os}}_{\text{D}} = \mu\sigma_z\tau_z \tag{25}$$

$$H^{\text{os,c}}_{\text{D}} = \lambda_1\sigma_z\tau_z + i\lambda_2\sigma_0\tau_0 \tag{26}$$

$$H^{\text{os}}_{\text{CII}} = \mu\rho_z\sigma_z\tau_0 + \lambda_1\rho_z\sigma_x\tau_0 + \rho_x\sigma_0\cdot(\lambda_2\tau_z + \lambda_3\tau_x) \tag{27}$$

$$\begin{aligned}H^{\text{os}}_{\text{DIII}} = {}& \mu\rho_z\sigma_0\tau_z + \lambda_1\rho_y\sigma_y\tau_0 + \lambda_2\rho_x\sigma_x\tau_x \\ &+ \lambda_3\rho_z\sigma_z\tau_y + \lambda_4\rho_x\sigma_y\tau_0 + \lambda_5\rho_y\sigma_y\tau_y\,,\end{aligned} \tag{28}$$

where the Pauli matrices $\sigma$ and $\tau$ act on the electron-hole and the angular momentum degrees of freedom respectively. In the doubled models we assign different parameter values in the two diagonal blocks.

The doubled $k$-space models have the following hopping terms:

$$H^{\text{hop}}_{\text{AIII}}(\mathbf{k}) = t_n\sigma_z\tau_z(k_x^2 + k_y^2) + (t1\sigma_z - t_2\sigma_y)\tau_x k_x + (t_1\sigma_z + t_2\sigma_x)\tau_y k_y \tag{29}$$

$$H^{\text{hop,c}}_{\text{AIII}}(\mathbf{k}) = (\beta_1\sigma_z + \beta_2\sigma_y)\tau_x k_x + (\beta_1\sigma_z + \beta_2^*\sigma_y)\tau_y k_y \tag{30}$$

$$H^{\text{hop}}_{\text{BDI}}(\mathbf{k}) = t_n\sigma_z\tau_z(k_x^2 + k_y^2) + t\sigma_z(\tau_x k_x + \sigma_z\tau_y k_y) \tag{31}$$

$$H^{\text{hop,c}}_{\text{BDI}}(\mathbf{k}) = o_1\sigma_z(\tau_x k_x + \tau_y k_y) + io_2\sigma_y(\tau_x k_x + \tau_y k_y) \tag{32}$$

$$H_{\mathrm{D}}^{\mathrm{hop}}(\mathbf{k}) = t_n \sigma_z \tau_z (k_x^2 + k_y^2) + t_1 \sigma_z (\tau_x k_x + \tau_y k_y)$$
$$+ t_2 \sigma_0 (-\tau_y k_x + \tau_x k_y) + d\sigma_x (\tau_x k_x + \tau_y k_y) \tag{33}$$

$$H_{\mathrm{D}}^{\mathrm{hop,c}}(\mathbf{k}) = io_1 \sigma_y (\tau_x k_x + \tau_y k_y) + o_2 \sigma_z (\tau_x k_x + \tau_y k_y)$$
$$+ o_3 \sigma_x (-\tau_y k_x + \tau_x k_y) + o_4 \sigma_0 (-\tau_y k_x + \tau_x k_y). \tag{34}$$

The k-space CII and DIII models have hopping terms of the form:

$$H_{\mathrm{CII}}^{\mathrm{hop}}(\mathbf{k}) = t_n \rho_z \sigma_z \tau_0 (k_x^2 + k_y^2)$$
$$+ t_1 (\rho_z \sigma_z \tau_z k_x + \rho_0 \sigma_0 \tau_x k_y) + t_2 (\rho_z \sigma_0 \tau_x k_x - \rho_0 \sigma_0 \tau_z k_y)$$
$$+ t_3 (\rho_x \sigma_0 \tau_0 k_x + \rho_y \sigma_z \tau_y k_y) + t_4 (\rho_x \sigma_x \tau_0 k_x + \rho_y \sigma_x \tau_y k_y) \tag{35}$$

$$H_{\mathrm{DIII}}^{\mathrm{hop}}(\mathbf{k}) = t_n \rho_z \sigma_0 \tau_z (k_x^2 + k_y^2)$$
$$+ d(\rho_0 \sigma_y \tau_x k_x + \rho_0 \sigma_0 \tau_y k_y) + t(-\rho_0 \sigma_x \tau_0 k_x + \rho_0 \sigma_z \tau_z k_y)$$
$$+ o_1 (\rho_y \sigma_z \tau_z k_x + \rho_y \sigma_0 \tau_0 k_y) + o_2 (\rho_x \sigma_0 \tau_y k_x + \rho_x \sigma_y \tau_0 k_y)$$
$$+ o_3 (\rho_x \sigma_z \tau_z k_x + \rho_x \sigma_x \tau_0 k_y) + o_4 (\rho_y \sigma_0 \tau_y k_x - \rho_y \sigma_y \tau_x k_y). \tag{36}$$

## E.2 Real space Hamiltonians

For the real-space models the onsite Hamiltonian are identical to the onsite terms found in the previous section.

The double model hopping Hamiltonians have the form:

$$H_{\mathrm{AIII}}^{\mathrm{hop}}(\hat{\mathbf{d}}) = t_n \sigma_z \tau_z + it\sigma_z (\tau_x d_x + \tau_y d_y) + id\sigma_y (\tau_x d_x + \tau_y d_y) \tag{37}$$

$$H_{\mathrm{AIII}}^{\mathrm{hop,c}}(\hat{\mathbf{d}}) = o_1 \sigma_z (i\tau_x d_x + i\tau_y d_y) + o_2 \sigma_0 (i\tau_y d_x + i\tau_x d_y)$$
$$+ o_3 \sigma_y (\tau_x d_x + \tau_y d_y) + o_4 \sigma_x (-i\tau_y d_x + i\tau_x d_y) \tag{38}$$

$$H_{\mathrm{BDI}}^{\mathrm{hop}}(\hat{\mathbf{d}}) = t_n \sigma_z \tau_z + it\sigma_z (\tau_x d_x + \tau_y d_y) \tag{39}$$

$$H_{\mathrm{BDI}}^{\mathrm{hop,c}}(\hat{\mathbf{d}}) = io_1 \sigma_z (\tau_x d_x + \tau_y d_y) + io_2 \sigma_y (\tau_x d_x - \tau_y d_y) \tag{40}$$

$$H_{\mathrm{D}}^{\mathrm{hop}}(\hat{\mathbf{d}}) = t_n \sigma_z \tau_z + it_1 \sigma_z (\tau_y d_x - \tau_x d_y) + it_2 \sigma_0 (\tau_x d_x + \tau_y d_y)$$
$$+ id\sigma_x (\tau_x d_x + \tau_y d_y) \tag{41}$$

$$H_{\mathrm{D}}^{\mathrm{hop,c}}(\hat{\mathbf{d}}) = io_1 \sigma_z (\tau_x d_x + \tau_y d_y) + io_2 \sigma_y (\tau_x d_x + \tau_y d_y)$$
$$+ io_3 \sigma_x (\tau_y d_x + \tau_x d_y) + io_4 \sigma_0 (\tau_y d_x - \tau_x d_y). \tag{42}$$

The 8-band CII and DIII models have hopping terms:

$$H_{\mathrm{CII}}^{\mathrm{hop}}(\hat{\mathbf{d}}) = t_n \rho_z \sigma_z \tau_0 + it_1 (\rho_z \sigma_0 \tau_z d_x + \rho_0 \sigma_0 \tau_x d_y) + it_2 (\rho_z \sigma_0 \tau_x d_x - \rho_0 \sigma_0 \tau_z d_y)$$
$$+ it_3 (\rho_x \sigma_z \tau_0 d_x + \rho_y \sigma_z \tau_y d_y) + it_4 (\rho_x \sigma_x \tau_0 d_x + \rho_y \sigma_x \tau_y d_y) \tag{43}$$

$$H_{\mathrm{DIII}}^{\mathrm{hop}}(\hat{\mathbf{d}}) = t_n \rho_z \sigma_0 \tau_z + id(\rho_0 \sigma_y \tau_x d_x + \rho_0 \sigma_0 \tau_y d_y) + it(-\rho_0 \sigma_x \tau_0 d_x + \rho_0 \sigma_z \tau_z d_y)$$
$$+ io_1 (\rho_y \sigma_z \tau_z d_x + \rho_y \sigma_0 \tau_0 d_y) + io_2 (\rho_x \sigma_0 \tau_y d_x + \rho_x \sigma_y \tau_0 d_y)$$
$$+ io_3 (\rho_x \sigma_z \tau_z d_x + \rho_x \sigma_x \tau_0 d_y) + io_4 (\rho_y \sigma_0 \tau_y d_x - \rho_y \sigma_y \tau_x d_y). \tag{44}$$

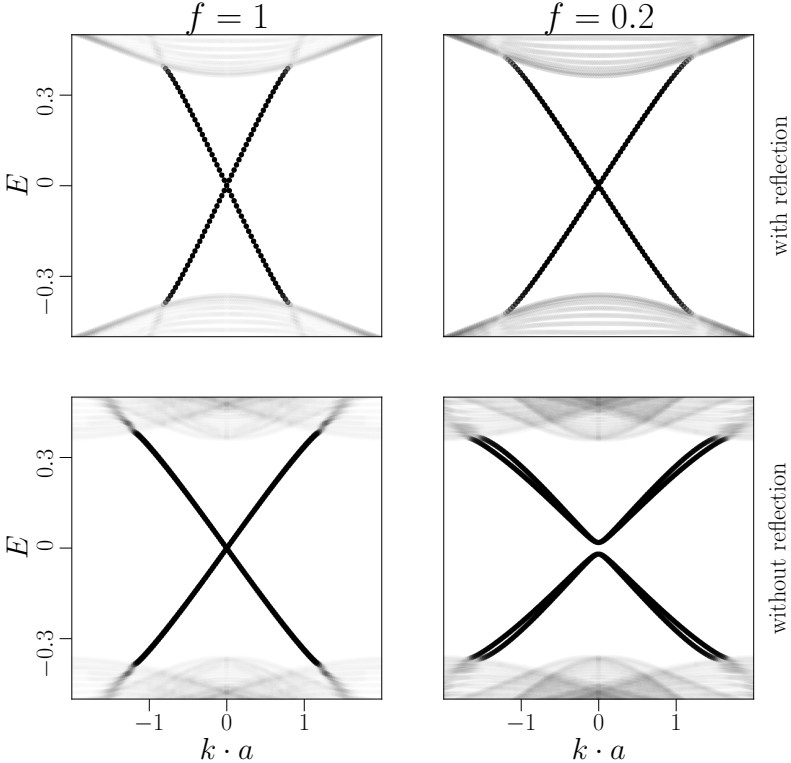

Figure 6: Band structures of the class D model on periodic crystal strips for different edge terminations and distance dependences. Top panels are bands along the reflection symmetric [1 0] edge, and bottom panels are bands along the reflection asymmetric [2 1] edge. Transparency of the dispersion bands is directly proportional to their participation ratio $\sum_i |\psi_i|^4$, where $\psi_i$ is the real space wavefunction of site $i$ of the system. Plot details in App. A.

## F  Removing additional symmetries of square lattice models

We find that because the nearest-neighbour square lattice is bipartite, it has inherent sublattice (chiral) symmetry that stabilizes an additional pair of counter-propagating edge modes at $k = \pi$. When studying models on the square lattice, we include second and third nearest-neighbour bonds to remove this chiral symmetry and the additional modes.

    We find that if every hopping decays the same way with the bond length, even the edges of a crystalline sample that break reflection behave like the edge of a fully isotropic continuum sample that has protected modes for every orientation close to $k = 0$. Hence without changing the symmetry properties we include a different decay constant in the prefactor for each term:

$$H^{\mathrm{hop}}(\mathbf{d}) = \sum_i e^{-f_i \cdot |\mathbf{d}|} \alpha_i H_i^{\mathrm{hop}}(\hat{\mathbf{d}}), \tag{45}$$

where $i$ runs over the linearly independent hopping terms [24] in $H^{\mathrm{hop}}(\hat{\mathbf{d}}) = \sum_i \alpha_i H_i^{\mathrm{hop}}(\hat{\mathbf{d}})$. Fig. 6 and Fig. 7 illustrate the importance of this consideration.

    The band structures of the chiral class models are all gapped for edge orientations that break reflection symmetry, as seen in Fig. 7. For the class AIII model, Fig. 7 shows that the case is similar to the class D crystal bands: the more general distance dependence (absence of a global prefactor related to the bond lengths before each of the hopping terms) is required to open the gap along reflection asymmetric edges. For the class BDI model, the reflection

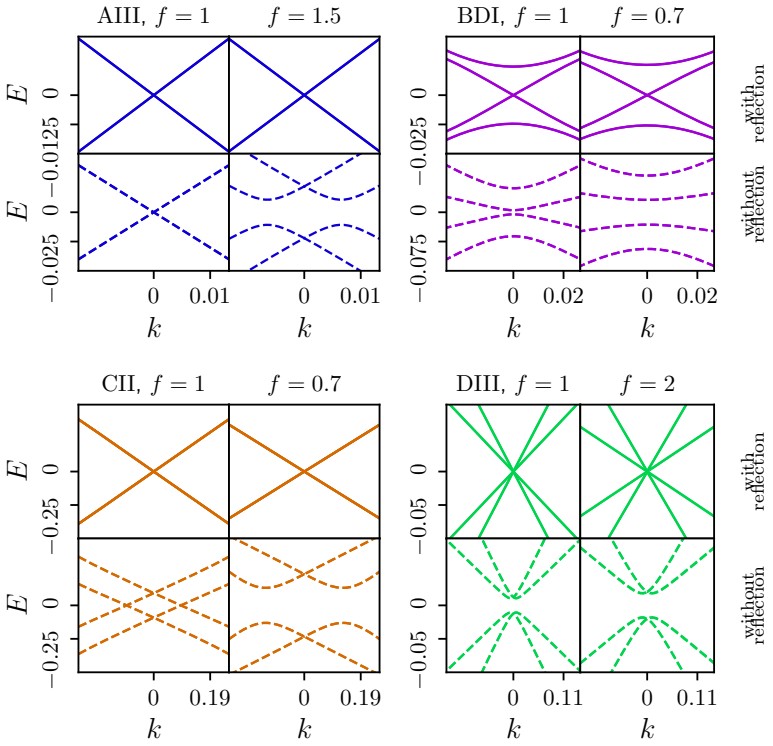

Figure 7: Band structures of the chiral models on periodic crystal strips for different edge terminations and hopping relations. Solid lines indicate bands recorded along the reflection symmetric [1 0] edge, and dashed lines along the reflection asymmetric [2 1] edge. The different hopping relations are distinguished by different values of $f$ from (45), see App. F.

asymmetric edges are gapped even without the more general distance dependence, as seen in Fig. 7, but it does increase the size of the gap. The situation is similar for the CII model, where the more general distance dependence of the hopping opens a gap only on reflection asymmetric edges.

## G  Transport scaling

Fig. 8 shows the transport scaling of the class D amorphous model without onsite disorder. The scaling arises from the intrinsic noise of the random graph. The bottom panel shows that we recover the relation $\sigma \propto \sqrt{L}$ for the standard deviation of $\alpha$. The conductance data in the inset shows that the noise due to the physical randomness of the amorphous system has a much weaker effect on localizing the modes compared to the noise originating from terms added to the model as in Fig. 3. The conductance relation $g \propto L^{-1/2}$ is not recoverable with the numerically accessible edge lengths, as it is only valid for $g \ll 1$.

## H  Bulk invariant for chiral classes

In this section we construct invariants classifying continuum and amorphous systems protected by continuous rotation and unitary reflection symmetry.

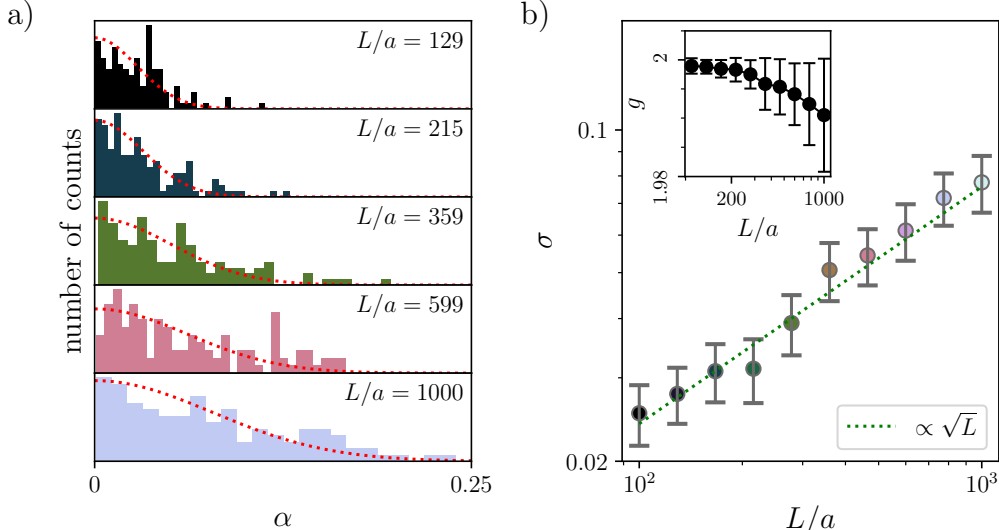

Figure 8: Critical transport scaling for the 8-band class D amorphous system without onsite disorder. Top panel: histogram of $\alpha$ for various system sizes $L$ from 59 different amorphous system realizations. In red: maximum likelihood estimate fit of a half-normal distribution to the data. Bottom panel: length dependence of $\sigma$, the square root of the scale factors of the half-normal fits. Inset: average conductance $g$ as a function of system size. The dashed line indicates the $L^{1/2}$ fit to the data.

## H.1  Classes AIII, BDI and CII

In the presence of chiral symmetry, the band-flattened Hamiltonian $Q(\mathbf{k})$ can be rearranged into two off-diagonal blocks in the basis where $\mathcal{C} = \tau_z$ [3,5]:

$$Q(\mathbf{k}) = \begin{pmatrix} 0 & q(\mathbf{k}) \\ q^\dagger(\mathbf{k}) & 0 \end{pmatrix}. \tag{46}$$

As $[S_z, \mathcal{C}] = 0$ we can simultaneously diagonalize the two operators and choose $S_z = s_z \tau_z$ where $s_z$ is diagonal. A mirror operator $U_M$ anticommutes with $S_z$ and we fix $U_M^2 = +\mathbb{1}$ in the following, this can always be achieved by choosing its overall phase. A mirror either commutes or anticommutes with $\mathcal{C}$, here we assume $[U_M, \mathcal{C}] = 0$ as we found in Sec. 3.1 that all symmetry groups protecting gapless edges have this property. In this case $U_M$ takes a block-diagonal form with diagonal blocks $m$ and $m'$, both of which square to $+\mathbb{1}$ and anticommute with $s_z$, guaranteeing that the spectrum of $s_z$ is symmetric. Because of this, $m$ (also $m'$) is only nonzero between opposite $s_z$ eigenvalues, an appropriately chosen block-diagonal basis transformation that preserves the form of $\mathcal{C}$ and $S_z$ makes it proportional to $\sigma_x$ in each $|s_z|$ sector. Hence there is always a basis where $m = m' = \sigma_x \otimes \mathbb{1}$ and the symmetry constraint is $m\, q(\mathbf{k})\, m^{-1} = q(R_M \mathbf{k})$.

This allows to decompose $q(\mathbf{k})$ into even/odd mirror sectors $q_\pm(\mathbf{k})$ with respect to a mirror operator that leaves $\mathbf{k}$ invariant [50], and to assign an individual winding number along a mirror-invariant line:

$$n_\pm = -\frac{1}{2\pi} \int_{-\infty}^{\infty} dk\, \frac{d}{dk} \arg \det q_\pm(k\hat{\mathbf{n}}), \tag{47}$$

where the sectors are with respect to the reflection operator with normal orthogonal to $\hat{\mathbf{n}}$. Due to the regularization of the Hamiltonian the integral is along a closed loop, hence quantized to integers, $n_\pm \in \mathbb{Z}$. The twofold rotation symmetry $C_2 = \exp\left(i\frac{\pi}{2}S_z\right)$ reverses $\mathbf{k}$ and for integer

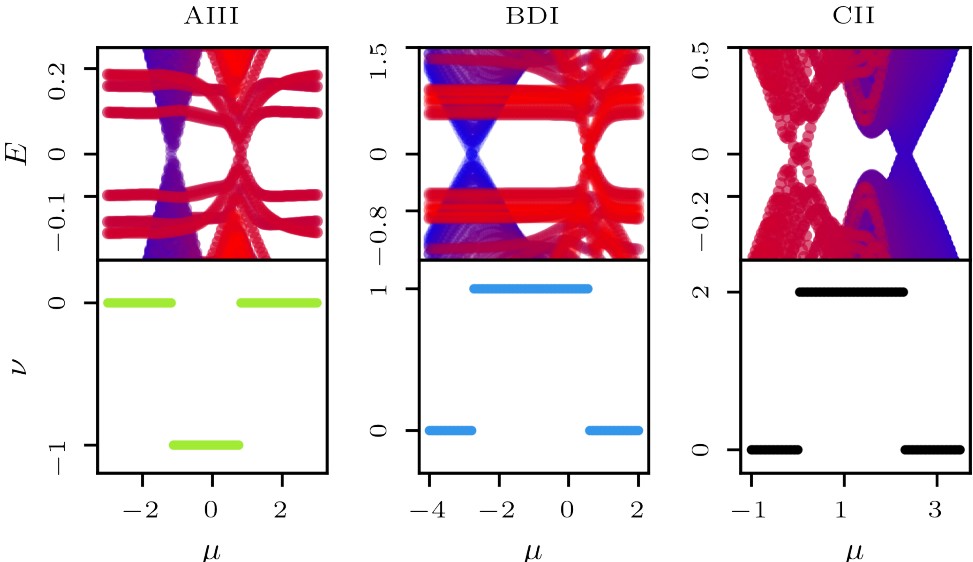

Figure 9: The bulk invariant $\nu$ as a function of the chemical potential $\mu$ calculated using the effective Hamiltonian for chiral models AIII, BDI, and CII. Top panels: the bulk spectra of the effective Hamiltonians. States at $k = 0$ are in red, states at $k = \infty$ are in blue, and states at intermediate $k$ are in varying shades of purple. Bottom panels: winding number invariants (50) obtained by dividing the integration space into 20 (AIII, CII) or 50 (BDI) segments.

or half-integer spin commutes or anticommutes with $U_M$ respectively. For the integer case this means for the winding numbers that $n_+ = n_- = 0$ making the invariant trivial, while in the half-integer case $n_+ = -n_-$ meaning that the total winding $n$ vanishes. So in the half-integer $S_z$ case we can select either one of the reflection-resolved windings to define a nontrivial topological invariant $n_M = \pm n_\pm$. As argued in Sec. 5.1 this implies the presence of $n_M$ zero modes in each mirror sector at $k = 0$ on any straight edge. In class CII time reversal symmetry imposes Kramers-degeneracy making $n_M$ even.

The winding number invariant we found for continuum systems is integer valued, suggesting that it is possible for the edge to host more than one pair of counter-propagating modes. In the presence of disorder, however, an even multiple of the minimum number of symmetry-allowed counter-propagating mode pairs always localizes [11]. In classes AIII and BDI (CII) this renders edges of systems with even $n_M$ ($n_M/2$) insulating, and those with odd $n_M$ ($n_M/2$) indistingushable through transport probes. Therefore, rather than the winding number $n_M \in \mathbb{Z}$ itself being our invariant for amorphous systems, we identify its parity $\nu_M \in \mathbb{Z}_2$ as the mirror invariant in classes AIII and BDI:

$$\nu_M = n_M \bmod 2\,, \tag{48}$$

and the parity of half of $n_M \in 2\mathbb{Z}$ in class CII:

$$\nu_M = \frac{n_M}{2} \bmod 2\,. \tag{49}$$

We calculate the $\mathbb{Z}_2$ invariant for the effective Hamiltonian of the amorphous models in all the chiral symmetry classes as the chemical potential $\mu$ is tuned across two topological phase transitions, the result is shown in Fig. 9. For the numerical calculation we discretize the

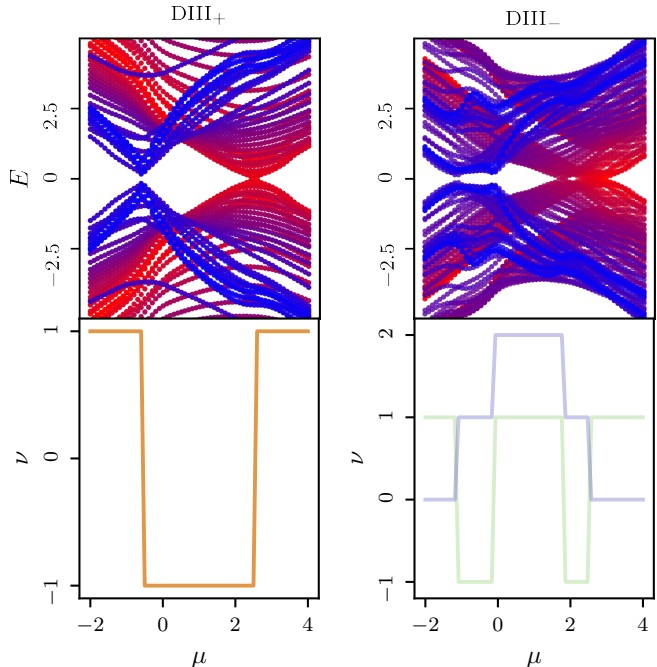

Figure 10: The bulk invariant $\nu$ as a function of the chemical potential $\mu$ calculated using the effective Hamiltonian for the DIII model in the commuting (DIII$_+$) and anti-commuting (DIII$_-$) cases. Top panels: the bulk spectra of the effective Hamiltonians. States at $k = 0$ are in red, states at $k = \infty$ are in blue, and states at intermediate $k$ are in varying shades of purple. Bottom panels: the DIII $\mathbb{Z}_2$ mirror-resolved strong invariant (53) shown for the DIII$_+$ case (in orange). For the DIII$_-$ case, the DIII $\mathbb{Z}_2$ strong invariant (51) is shown in green, and the winding number invariant is shown in purple.

integral in equation (47) as

$$ n_M \approx -\frac{1}{2\pi} \sum_i \operatorname{Im} \log\left( \frac{\det q_\pm(k_{i+1}\hat{\mathbf{n}})}{\det q_\pm(k_i\hat{\mathbf{n}})} \right), \tag{50}$$

where $k_i$ is a discrete set of $k$-values in increasing order and with cyclic indexing. To address numerical integration to infinity, we choose the parametrization $k = \tan(\phi/2)$ where $\phi$ corresponds to the latitude in stereographic projection ranging from $-\pi$ to $\pi$. We use 10 evenly spaced values for $\phi$ in the numerical calculations, we show the results in Fig. 9.

## H.2  Class DIII

In this section we show that the above invariant, while well defined in classes DIII$_\pm$, in class DIII$_+$ it always vanishes, and in class DIII$_-$ its parity is determined by the strong $\mathbb{Z}_2$ invariant of class DIII. For class DIII$_+$ we introduce a different $\mathbb{Z}_2$ invariant that is independent of the strong invariant.

We start by deriving general symmetry constraints. We choose the onsite symmetry representation as $\mathcal{C} = \tau_z$, $\mathcal{T} = \tau_y \mathcal{K}$ and $\mathcal{P} = \tau_x \mathcal{K}$, in this basis the Hamiltonian has the off-diagonal form of (46) with $q(\mathbf{k}) = -q(-\mathbf{k})^T$ [5]. This form of the symmetries is invariant under basis transformations of the block-diagonal form $\operatorname{diag}(u, u^*)$ which allows to bring spin operator to the diagonal form $S_z = \operatorname{diag}(s_z, -s_z)$. For half-integer $S_z$ the combination $C_2 \mathcal{T}$ leaves $\mathbf{k}$ invariant and acts as $\sigma_z q(\mathbf{k}) \sigma_z = q(\mathbf{k})^T$. We find for the mirror operator that

it takes a block-diagonal form $M = \text{diag}(m, \pm m^*)$ where the $\pm$ stands for the commuting ($[U_M, \mathcal{P}] = [U_M, \mathcal{T}] = 0$) and anticommuting ($\{U_M, \mathcal{P}\} = \{U_M, \mathcal{T}\} = 0$) case. As $m$ anticommutes with $s_z$ it is only nonzero in the off-diagonal blocks connecting opposite spin eigenvalues. In a single $|s_z| \neq 0$ sector $s_z \propto \sigma_z$, and $m$ has off-diagonal blocks $\mu$ and $\mu^\dagger$, these can be diagonalized by a basis transformation that in this sector acts as $\text{diag}(\mathbb{1}, \mu)$. For class DIII$_+$ (DIII$_-$) we bring the reflection operator to the form $m = \sigma_x$ ($m = \sigma_y$) which imposes the constraint $\sigma_x q(\mathbf{k}) \sigma_x = q(\mathbf{k})$ ($\sigma_y q(\mathbf{k}) \sigma_y = q(\mathbf{k})$). We transform to a basis where $m = \sigma_z$ using $u = \exp(i\pi/2\sigma_y)$ ($u = \exp(i\pi/2\sigma_x)$), in this basis $q_\pm$ are the diagonal (off-diagonal) blocks of $q$ and the $C_2 \mathcal{T}$ constraint reads $q_+(\mathbf{k}) = q_-(\mathbf{k})^T$ ($q_\pm(\mathbf{k}) = q_\pm(\mathbf{k})^T$). In DIII$_+$ this implies $\det q_+(\mathbf{k}) = \det q_-(\mathbf{k})^T$, meaning that the winding is the same in both sectors, however, the total winding always vanishes in class DIII, so the reflection-resolved windings also vanish.

We write the 1D class DIII $\mathbb{Z}_2$ strong invariant [5] adapted to the compactified $k$-space as

$$\nu = \frac{\text{Pf}\, q(\boldsymbol{\infty})}{\text{Pf}\, q(\mathbf{0})} \exp\left(-\frac{i}{2} \int_0^\infty dk \frac{d}{dk} \arg \det q(k\hat{\mathbf{n}})\right). \tag{51}$$

This is also the strong 2D invariant, as the $k$-space sphere only has two time-reversal invariant momenta at $\mathbf{k} = \mathbf{0}$ and $\boldsymbol{\infty}$. In class DIII$_-$ $q$ has off-diagonal blocks $q_\pm$ and $q_+(\mathbf{k}) = -q_-(\mathbf{k})^T$ for $\mathbf{k} = \mathbf{0}$ and $\boldsymbol{\infty}$, meaning $\text{pf}\, q(\mathbf{k}) = (-1)^{n(n-1)/2} \det q_+(\mathbf{k})$ where $n$ is the size of a reflection block. Using that $q_\pm(\mathbf{k}) = -q_\mp(-\mathbf{k})^T$ for all $\mathbf{k}$, adding and subtracting the winding $i\pi n_+$ in the exponential, and noting that the winding of the phase of the determinant between two points can only differ from the difference in the phases at the endpoints by a multiple of $2\pi$, we find

$$\nu = e^{i\pi n_M}, \tag{52}$$

showing that the parity of $n_M$, hence the protection of gapless edges in the presence of disorder, is given by the strong invariant.

We define an invariant for class DIII$_+$ in terms of the reflection-resolved class DIII $\mathbb{Z}_2$ invariant:

$$\nu_\pm = \frac{\text{pf}\, q_\pm(\boldsymbol{\infty})}{\text{pf}\, q_\pm(\mathbf{0})} \exp\left(-\frac{i}{2} \int_0^\infty dk \frac{d}{dk} \arg \det q_\pm(k\hat{\mathbf{n}})\right). \tag{53}$$

As follows from the relations derived above, the invariant is the same for both sectors and we define the mirror invariant as $\nu_M = \nu_\pm$. This also shows that in class DIII$_+$ the strong invariant is always trivial in the presence of reflection symmetry.

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
