# Peer review of "Amorphous topological phases protected by continuous rotation symmetry"

_SciPost Physics, doi:SciPost Phys. 11, 022 (2021)_

## Round 1 · Referee Report · Anonymous (Referee 3) · 2021-6-4

Report

I thank the authors’ clarification for some of my questions. Hope that comments 3 and 4 can be addressed before my recommendation. Furthermore, I would like to comment that the structural disorder is randomly added around a cubic lattice configuration in Ref. [21] and when the disorder is sufficiently strong, e.g., W=L (system size), the configuration corresponds to an independent uniformly distributed sites without any correlation between sites, similar to the hard disk amorphous structures employed in the paper.
  • validity: high
  • significance: high
  • originality: high
  • clarity: good
  • formatting: good
  • grammar: good

Author:  Helene Spring  on 2021-06-23  [id 1516]

(in reply to Report 1 on 2021-06-04)

Dear referee,

We apologize for not being fully explicit in answering the remarks 3 and 4. We have incorporated the appropriate amendments in the revised manuscript, but did not list the responses separately, which we now do below.

  1. In the second paragraph of subsection A. Continuum models in Sec. V, the authors consider a class D Hamiltonian to explain the classification based on the reflection symmetry and particle-hole symmetry. I think that it is only when the Chern number is zero that one needs to classify the phase by considering the reflection symmetry. Can the authors clarify this point?

This is indeed correct: we focus on phases that are topological exclusively due to the spatial symmetry. Phases with a finite Chern number stay protected also when all spatial symmetries are broken, and are therefore not relevant to our analysis. In the amended manuscript we emphasize this by stating that the Chern number is zero in the presence of reflection symmetry.

  1. In Table II, the authors display the classification for continuum and amorphous systems. I think the authors should explain in more detail why the classification is different for amorphous and continuum systems. For example, in the class AIII, why does the classification become $\mathbb{Z}_2$ in amorphous systems compared with the $\mathbb{Z}$ classification in continuum systems.

In this distinction we rely on the previous analysis of stability of gapless edge states against disorder, in presence of ensemble symmetries (Ref. 11 of the manuscript). We have also added a reference to App. H from the caption of Table II, where this difference is explained in detail.

The referee also makes a remark that displacing atoms from their lattice positions by the system size results in uncorrelated atomic positions. While this is correct, a random displacement of sites of a cubic lattice within a cube of size $W$ results not only in correlations, but also in periodic density variation unless $W$ is an integer multiple of the unit cell size. Due to this subtlety, we prefer to not recommend the readers to use random displacements to generate amorphous media with exponentially decaying spatial correlations. We believe that the amended manuscript makes this sufficiently clear by using the formulation "These conditions require care to satisfy and cannot be fulfilled by gradually moving sites from their crystalline positions".

Best regards, Helene Spring, Anton Akhmerov and Daniel Varjas

---

## Round 1 · Referee Report · Anonymous (Referee 1) · 2021-6-29

Report

In their response and the new version, the authors address my questions and suggestions and have implemented a number of modifications in the manuscript. Based on these, I am happy to recommend publication of the manuscript in the present form.

---

## Round 1 · Author Response

We thank the referees for the time and effort in reviewing our manuscript. Below we address the technical questions posed in their reports and the changes we implemented in view of our resubmission.

---

## Round 1 · List of Changes

Both referees asked for a specific definition of amorphous matter — a request that we address in the newly added first paragraph of section II. Amorphous matter must be fully invariant under all Euclidean transformations and have no long-range correlations, unlike in the reference suggested by Referee 2, which we now use to clarify the distinction. Turning to the question of Referee 1 about whether residual correlations would open small gaps, we indeed expect that models lacking exact rotational symmetry would have boundary-mode localization at the surfaces that do not respect ensemble reflection symmetry.

The second referee also asked whether the disorder must respect certain symmetries exactly. This is indeed the case, as we now explain in the fourth paragraph of section III, and as was studied in PRB 89 155424. This reference also demonstrates that even topological invariants lead to edge mode localization in the presence of disorder. We rely on this observation to deduce the difference in classification between the amorphous and continuum models.

Following the suggestion of Referee 1, we have specified the expected transport signature of our amorphous systems in the abstract, to make the findings of the paper clearer from the start.

Regarding the remark that all amorphous systems are gapless in principle (point 5 of report 1) we do not think that this is generally true, and that it is possible to have insulating amorphous structures. For example, an amorphous atomic insulator, that consists of disconnected atoms with a gapped spectrum, will also feature a hard gap in the thermodynamic limit, a feature that is robust against adding weak hopping between the atoms. As such, we do not believe that it is a numerical error that we do not see a finite local density of states in the bulk gap. On the other hand, our invariants are likely still valid even if there is a finite DOS of localised states in the gap, however, we consider this question beyond the scope of the current manuscript.

---

## Editorial Decision

published